



# Monitoring Glacier Calving using Underwater Sound

Jarosław Tęgowski[1,*], Oskar Glowacki[2,*], Michał Ciepły[3], Małgorzata Błaszczyk[3], Jacek Jania[3],
Mateusz Moskalik[2], Philippe Blondel[4], and Grant B. Deane[5]

[1]Institute of Oceanography, University of Gdansk, Gdynia, Poland
[2]Institute of Geophysics, Polish Academy of Sciences, Warsaw, Poland
[3]Faculty of Natural Sciences, Institute of Earth Sciences, University of Silesia in Katowice, Sosnowiec, Poland
[4]Department of Physics, University of Bath, Bath, UK
[5]Scripps Institution of Oceanography, UCSD, La Jolla, California, USA
[*]These authors contributed equally to this work.

**Correspondence:** Oskar Glowacki (oglowacki@igf.edu.pl), Grant B. Deane (gdeane@ucsd.edu)

**Abstract.** Climate shifts are particularly conspicuous in the Arctic. Satellite and terrestrial observations show significant increases in the melting and breakup of Arctic tidewater glaciers and their influence on sea level rise. Increasing melt rates are creating an urgency to better understand the link between atmospheric and oceanic conditions and glacier frontal ablation through iceberg calving and melting. Elucidating this link requires a combination of short and long-time scale measurements of terminus activity. Recent work has demonstrated the potential of using underwater sound to quantify the time and scale of calving events to yield integrated estimates of ice mass loss (Glowacki and Deane, 2020). Here, we present estimates of subaerial calving flux using underwater sound recorded at Hansbreen, Svalbard in September 2013 combined with an algorithm for the automatic detection of calving events. The method is compared with ice calving volumes estimated from geodetic measurements of the movement of the glacier terminus and an analysis of satellite images. The total volume of above-water calving during the 26 days of acoustical observation is estimated to be $1.7 \pm 0.7 \times 10^7 \, \mathrm{m}^3$, whereas the subaerial calving flux estimated by traditional methods is $7 \pm 2 \times 10^6 \, \mathrm{m}^3$. The results suggest that passive cryoacoustics is a viable technique for long-term monitoring of mass loss from marine-terminating glaciers.

## 1 Introduction

The loss in the mass of the Greenland ice sheet, which increased from $41 \pm 17$ Gt yr$^{-1}$ in 1990–2000 to $286 \pm 20$ Gt yr$^{-1}$ in 2010-2018 (Mouginot et al., 2019), is of great concern. Greenland's contribution to sea level rise by the end of the 21st century is estimated to be $90 \pm 50$ and $32 \pm 17$ mm for RCP8.5 and RCP2.6 greenhouse gas concentration scenarios, respectively (Goelzer et al., 2020); these estimates exclude peripheral glaciers and ice caps. Moreover, recent studies have shown that the accelerated break-up and melting of Antarctica – the largest reservoir of fresh water on Earth – will be responsible for a sea level rise of at least 0.5 cm yr$^{-1}$ by 2100 (Paolo et al., 2015; DeConto et al., 2021). Despite these concerning numbers, the loss of land-based ice in the coming decades could be under-predicted. For example, Greenland and Antarctic ice sheets likely have tipping points at around $1.5 - 2.0$°C of a mean global temperature increase compared to the pre-industrial era (Pattyn et al., 2018; Pattyn and Morlighem, 2020; Noël et al., 2021). The severe consequences of sea level rise and current uncertainties in





models of glacial retreat suggest that long-term monitoring of glacial stability is both important and urgent (Straneo et al., 2019).

There are many challenges to the long-term monitoring of glaciers and ice shelves. Polar regions are remote and plunged into darkness for up to 6 months a year. Moreover, capturing the episodic character of ice breakup usually requires sub-second temporal resolution over year-long time scales. Despite a suite of current techniques, including satellite technology (Smith et al., 2020; Podgórski and Pętlicki, 2020), camera observations (Medrzycka et al., 2016; How et al., 2019), terrestrial laser scanning (Pętlicki and Kinnard, 2016; Podgórski et al., 2018), ground-based radar imaging (Xie et al., 2018; Walter et al.,

2020), sonar scans (Sugiyama et al., 2019; Sutherland et al., 2019), and seismic records (Köhler et al., 2016; Podolskiy and Walter, 2016), continuous and long-term observations of the scale and rapidity of ice loss in logistically challenging polar regions are still urgently required (Straneo et al., 2013).

    There are 3 major processes driving mass loss from marine-terminating glaciers: surface melting, submarine melting, and calving, which is the mechanical loss of ice from the edges of glaciers or ice shelves (Benn et al., 2007b). Calving and

submarine melting - the two mechanisms of frontal ablation - are very difficult to quantify separately (Truffer and Motyka, 2016). Calving is a complex process that depends on a broad range of factors, including but not limited to (i) changes in glacier surface velocity along the flow line (longitudinal stretching), (ii) submarine melting of the terminus (undercutting), and (iii) geometry of the glacier-bay system (van der Veen, 2002; Benn et al., 2007b). This complexity makes the formulation of a general model of how calving changes in response to warming conditions a challenging task that requires long-term,

temporally-resolved data sets (e.g., van der Veen, 2002; Benn et al., 2007a).

    Passive cryoacoustics, the use of ambient sounds to study ice-ocean interactions in polar regions, has the potential to provide the long-term, temporally-resolved time series required for calving model development and monitoring the long-term stability of glaciers and ice sheets (see discussion in Glowacki and Deane, 2020), provided that the geophysical signals of relevance can be recovered from the properties of the sound. The field of ambient noise oceanography is well-established and has been

used to recover signals such as wind speed and precipitation, for example. The use of passive acoustics in polar regions is a relatively new field, still under development (Glowacki et al., 2015; Pettit et al., 2015; Deane et al., 2019; Yun et al., 2021; Podolskiy et al., 2022).

    A recent milestone has been the demonstration that the mass of a calving iceberg can be estimated from its underwater impact noise with the sea surface (Glowacki and Deane, 2020). The method exploits the observed power law relationship

between the kinetic energy of a falling ice block immediately prior to impact and the resulting sound energy produced. The kinetic energy of an impacting block can be estimated from the sound energy radiated provided that propagation losses between the block and the recording location can be estimated. Because the drop height of icebergs is limited by the vertical extent of the terminus, the estimate of impact energy can then be converted into a block mass using what Glowacki and Deane (2020) call a "mass-weighted average drop height", $\hat{h}$.

Here we demonstrate the use of passive cryoacoustics to monitor the subaerial calving flux from a tidewater glacier in southwestern Spitsbergen over 26 days of continuous measurements using algorithms for the automatic detection of calving





events and the removal of contaminating signals, such as iceberg disintegration. Acoustic estimates of the ice volume loss are compared with concurrent camera and satellite observations.

## 2 Study Site

The site for this study is Hansbreen, a retreating, grounded, polythermal tidewater glacier terminating in Hornsund Fjord, Svalbard. Its calving mode, believed to be driven by melt undercutting and many relatively small calving events from above the water surface (Pętlicki et al., 2015), provides a challenging test case for camera and satellite estimates of mass flux.

A schematic of the study site is shown in Fig. 1. Hansbreen covers an area of around 54 km$^2$, is more than 15 km long (Błaszczyk et al., 2013), and has a 1.5 km-wide active calving front with an average height of around 40 m (Błaszczyk et al.,
2009, 2021), which is approximately 30–40% of the total ice thickness. The glacier surface flow is dominated by basal motion in the ablation area (Vieli et al., 2004) and the mean annual flow velocity near the terminus and its calving flux is estimated to be 177 m y$^{-1}$ and 35×10$^6$ m$^3$y$^{-1}$, respectively (Błaszczyk et al., 2019).

Both glacial behavior (Pętlicki et al., 2015; Błaszczyk et al., 2021) and the propagation of calving sounds (Glowacki et al., 2016) are sensitive to the space and time-varying thermohaline structure of water masses in the bay, which were characterized
with CTD casts. The water temperature and salinity in the center of the bay ranged from -1.8 °C to more than 2.0 °C and from 30 PSU to almost 35 PSU during 2015 and 2016 (Moskalik et al., 2018). Sound propagation also depends on water depth, which drops to almost 100 m along a transect parallel to the glacier terminus (see Fig. 1A). The morphology of the bay varies greatly, with numerous moraines and flat areas created during the retreat of Hansbreen (Moskalik et al., 2018).

## 3 Automatic Detection of Calving Events

Glowacki and Deane (2020) have shown that the mass of individual icebergs can be estimated from the sound they produce when they impact the sea surface. In that study, 169 subaerial calving events from Hansbreen were manually identified in time-lapse photography and simultaneous recordings of the ambient sound, both made roughly 1 km from the glacier terminus. Despite significant variability in the total sound energy produced by an impact event for icebergs of comparable size, Monte Carlo simulations of ensemble time-series show that accurate estimates of subaerial calving flux can be made if sufficient
events are averaged (40 events for 20% standard error for Hansbreen).

The manual identification of calving events using time-lapse photography and underwater sound is accurate but too labor intensive to be seriously considered for long-term data sets. This issue is addressed here by developing algorithms for the automatic detection of ice impact noise in the ambient sound recordings and rejection of interfering events, such as the disintegration of icebergs in the bay to obtain accurate estimates of ice loss from the terminus of Hansbreen due to calving.
The automatic detection of calving events is based on the observation that the ambient sound field in a glacial bay is composed of two distinct categories of sound sources: 1. sources that radiate noise with statistical properties that are stationary on the time-scale of a calving event (such as melting glacier ice, for example) and 2. transient sources, like calving, superposed



on this stationary background. During periods when noise in the bay is dominated by ice melting and calving, the signal during a 10-minute data segment has the form of a wandering baseline punctuated at random intervals by hill-shaped increases as
calving events occur. The level and noisiness of the baseline varies over time due to changes in the rate of ice melting on the glacier terminus, proximity to the hydrophone of icebergs drifting in the bay, and variable acoustic propagation through the bay. These varying conditions are unpredictable and require the selection of a baseline level and event detection threshold that is adaptable.

The signal processing scheme for event detection works as follows. A 10-minute segment of ambient sound is analyzed into
a spectrogram. The sound power, $P_b$, between a lower and upper frequency band, $f_0 = 30$ Hz to $f_1 = 100$ Hz is then computed using

$$P_b = B(\Omega) \otimes \left[ M(\Theta) \otimes \int_{f_0}^{f_1} P_{xx} df \right], \tag{1}$$

where $P_{xx}$ is the power spectral density estimate for the noise segment and the operators $M(\Theta)$ and $B(\Omega)$ are the filters described below. The selection of frequency limits is motivated by previous analyses of the calving noise presented in Glowacki
and Deane (2020) and Glowacki (2020) (see Appendix A: Materials and Methods for more details). The integral in equation 1 is calculated using the trapezoidal rule. The filter $M(\Theta)$ is a median filter of order $\Theta = 7$ and is useful for eliminating loud, impulsive noise sources that are too short to be calving noise. A boxcar filter $B(\Omega)$ of length $\Omega = 32$ points is applied to further reduce the effects of impulsive noise sources and random variations in noise power.

A baseline power for the noise in the absence of calving, $P_{base}$, is taken to be the median of the probability density distri-
bution of $P_b$. A threshold of detection, $P_{thres}$, is then selected to be 1.5 times the largest value of $P_b$ with relative likelihood

$$\phi(P_{thres}) > \phi(P_{base}) e^{-\beta}, \tag{2}$$

where $\beta = 5$. This constant plays a central role in detector performance and is referred to as the 'detection factor' (see Appendix A: Materials and Methods).
Calving events are identified as continuous periods of time when $P_b$ exceeds $P_{thres}$. The detection algorithm is based on the idea that calving events are relatively rare within 10-minute intervals and periods of time containing many successive events will drive the algorithm to select only the most energetic of them (see Appendix A: Materials and Methods).

An example of the detector output is shown in Fig. 2. Figure 2A shows 1 minute of ambient sound energy containing a calving event, presented in dB as a function of frequency and time. Figure 2B shows the output from the band-pass and
median filters, the baseline level, the threshold level, and detection of the calving event. The detector was run on the 26 days of recordings, resulting in the detection of 4258 events (see Appendix A: Materials and Methods for a discussion of how sensitive the algorithm is to the choice of detection factor). The primary output of the event detector are the event start and end times, $t_0$ and $t_1$.





## 4    Interfering Event Removal

The problem remains of identifying events that were detected as calving but which are not. Visual inspection of noise spectrograms combined with listening to the corresponding sound recordings revealed two major sources of interfering sounds: (1) the disintegration of icebergs close to the acoustic buoy, which sounds like a calving event because the iceberg sheds blocks of ice into the water as it disintegrates, and (2) water movement around the hydrophone (flow noise). Other interfering events include calving from the terminus with highly-energetic contacts between the falling iceberg and the terminus, icebergs collid-

ing with other ice pieces that occupy the sea surface along the terminus, and calving events involving ice block rotation in the air, resulting in large contact area between the iceberg and sea surface leading to atypical sound production. Although all these events represent a challenge for the technique, the most concerning are the flow noise and disintegrating icebergs, which can appear to be very large calving events. The interfering events must be removed from the calving inventory before the ice mass loss is estimated. To deal with the misclassification problem, we take two steps.

First, we take advantage of the fact that submarine melting of glacier ice also generates underwater noise due to the impulsive release of pressurized air bubbles into the water (Urick, 1971; Deane et al., 2014; Pettit et al., 2015). Glowacki et al. (2018) have demonstrated that the melt noise has an $\alpha$-stable distribution. Importantly, the characteristic exponent of the $\alpha$-stable distribution indicates proximity to a melting source. When the melting iceberg is close to the receiver, the underwater noise at frequencies 1—10 kHz is more impulsive and the exponent $\alpha$ deviates from 2. Here, we assume the following: if $\alpha < 1.95$ for

the noise segment immediately preceding the calving noise, there is a high likelihood that the corresponding event detected by the automated algorithm is in fact an iceberg disintegrating close to the hydrophone. Such events are discarded. In this case, 875 events were removed from the inventory.

Second, the noise segments are occasionally contaminated with flow noise, which originates from water turbulence advected past the hydrophone and is most pronounced at frequencies below $\sim 50$ Hz. An overlap in frequency between the calving

detector band (30–100 Hz) and the flow noise (2–50 Hz) can result in false classifications of the episodes of intense water flow around the hydrophone as calving events. To deal with this issue, we discard all events for which the ratio between the average power spectral density at 2–30 Hz and 50–1000 Hz is higher than or equal to 1. This procedure is based on the observation that calving noise extends into the 50–1000 Hz band, whereas flow noise is most pronounced at frequencies below 30 Hz. Using these criteria, 328 misclassifications were identified and manually validated by listening to the selected noise segments and

visual inspections of spectrograms. In total, 1128 events were discarded due to the fact that 75 events were identified as both flow noise and iceberg disintegration.

One issue remains, which is the fact that some fraction of the event signals found come from submarine calving events. Submarine calving events are a problem because the parameters in the power law relationship between the energy of noise production and kinetic energy of block impact are for subaerial events and will not apply to submarine events (Glowacki and

Deane, 2020). Thus any submarine events included in the calving inventory will not have their volume calculated correctly. Moreover, even if their volume were calculated correctly, they are not detected by the camera observations for this study.



The total number of submarine calving events in the present inventory is not expected to be large. Reported statistics of submarine calving from other study sites show that such events are typically much less common than subaerial calving (Minowa et al., 2018; Köhler et al., 2019; How et al., 2019). Glowacki (2022) estimated that submarine calving events at Hansbreen

are 8–10 times less frequent than subaerial events. It must be noted, however, that the low rate of occurrence of submarine calving is offset by the fact that submarine events often produce the biggest icebergs (Warren et al., 1995; Motyka, 1997; O'Neel et al., 2007; Glowacki, 2022). Consequently, attributing this component of the total calving flux to subaerial events may introduce an error disproportionately large relative to their number. In response to this problem, a new semi-automatic method for distinguishing submarine and subaerial calving events was recently proposed (Glowacki, 2022). However, there

is no way at the present time to classify calving styles from sound recordings using fully automated methods. The time and frequency structure of the underwater noise from submarine and subaerial calving differs significantly (Glowacki, 2022); for example, the underwater noise from submarine calving typically has longer duration, with individual sound-source mechanisms separated by the quiescent periods. As a result, a universal calving detector should (i) take into account the differences in time and frequency structure of the calving noise and (ii) be validated with high-frequency time-lapse images synchronized with

long-term acoustic recordings; this task is beyond the scope of the present study. Keeping this limitation in mind, the section that follows will explain how the subaerial calving flux can be estimated acoustically, with an assumption that all detected events are subaerial.

## 5   Estimating Ice Volume Loss

Following event detection, the next step is to compute the total noise energy radiated by the calving event at a standard reference

distance of 1 m, $E_{ac,imp}$. This step requires the calculation of the time and frequency integrated energy of the calving noise. The energy of the impact sound radiated by a falling ice block can be expressed in terms of the observed sound energy by accounting for propagation effects and the addition of background noise using:

$$E_{ac,imp} = \frac{4\pi}{\rho_w c_w} \int_{f_0}^{f_1} \left[ \left( \int_{t_0}^{t_0+\Delta t} P_{xx}\,dt - \int_{t_0-\Delta t}^{t_0} P_{xx}\,dt \right) \times 10^{\frac{-TL(f)}{10}} \right] df, \tag{3}$$

where $\rho_w$ is the water density, $c_w$ is the sound speed, $f_0 = 30$ Hz and $f_1 = 100$ Hz are lower and upper frequency limits, $t_0$

is the start time of the calving event determined from the event detector, $\Delta t = t_1 - t_0$ is the calving signal duration, where $t_1$ is the end time of the calving signal, $TL$ is the frequency-dependent transmission loss between the hydrophone and the point of ice block impact (in decibels). The acoustic signal is integrated over the surface area of a unit sphere ($4\pi$) to obtain total noise energy in joules. The two integrals over time, $\int_{t_0}^{t_0+\Delta t} P_{xx}\,dt$ and $\int_{t_0-\Delta t}^{t_0} P_{xx}\,dt$, are the frequency spectral densities of the calving noise plus background noise, and background noise alone, respectively. The transmission loss is calculated using

the standard numerical code RAM (Collins, 1993) (see Appendix A: Materials and Methods for further details).



In the next step, the kinetic energy of the falling block of ice, $E_{imp}$, is calculated from the total impact sound energy radiated into the bay, $E_{ac,imp}$, using the power law relationship:

$$E_{imp} = \gamma E_{ac,imp}^{\kappa} + \zeta, \tag{4}$$

where $\gamma = 1.45 \times 10^9$, $\kappa = 0.18$ and $\zeta = -2.66 \times 10^9$ are constants determined using the calibration dataset collected in 2016 (Glowacki and Deane (2020); see Appendix A: Materials and Methods for details). Finally, the iceberg volume is calculated as

$$V = \frac{M}{\rho_i} = \frac{E_{imp}}{\rho_i g \hat{h}}, \tag{5}$$

where $M$ is the iceberg mass, $\rho_i = 917$ kg m$^{-3}$ is assumed ice density, $g = 9.81$ m s$^{-2}$ is the acceleration due to gravity and $\hat{h} = 20.7$ m is the mass-weighted average drop height estimated for the terminus of Hansbreen (Glowacki and Deane, 2020).

## 6  Results

Figure 3 shows a comparison of two time series: (i) subaerial calving flux derived from iceberg impact noise and (ii) terminus area loss estimated from camera observations over the 26 days of analysis (see Appendix A: Materials and Methods). The ice volume loss was calculated every 3 hours, which is the sampling time of the camera system. Both data streams show variability between samples and between each other. This is to be expected; Glowacki and Deane (2020) demonstrated that the acoustic method is accurate if sufficient calving events are averaged but significant variability is observed in the signal between blocks of comparable volume. Moreover, camera observations are sensitive not only to weather conditions but also to the location of calving events due to the irregular shape of the terminus and oblique angle of view (see Fig. 1B). However, there is a clear relationship between the acoustic measurements of subaerial calving fluxes and the time-lapse observation of the terminus area loss, despite the limitations of both methods. The Pearson's correlation coefficient between the two data streams smoothed with a 1-day running average is 0.6.

The question now is: how do the acoustic measurements of the total subaerial calving flux compare to results obtained with more traditional methods? The cumulative subaerial ice volume loss over the 26 days of observation is $1.7 \pm 0.7 \times 10^7$ m$^3$ from the acoustic measurements and $7 \pm 2 \times 10^6$ m$^3$ from satellite data combined with stake measurements of the glacier velocity. The discrepancy may be due to several reasons. First, any submarine calving events are treated as subaerial in the acoustic analysis. Consequently, the ice volume loss from acoustics is almost certainly overestimated. This issue has been addressed recently by Glowacki (2022) and can be incorporated into a long-term analysis once a method is developed for the automatic detection of submarine calving. Second, the power law relationship in Eq. 4 is based on the calibration dataset that does not cover the full range of impact noise energies observed in 2013 data (see Fig. A2 and discussion in Appendix A: Materials and Methods). Third, the location of the hydrophone – far from the terminus and in shallow water – was unfavorable because the propagation conditions of the calving noise are very different for different segments of the terminus (see the variable bathymetry along the black dashed lines in Fig. 1A), which has a large effect on the transmission losses of low-frequency sounds in a glacial bay



(Glowacki and Deane, 2020). Finally, stake measurements of the ice velocity that were used to calculate subaerial calving flux from satellite images are from August 2013, which is a month before the acoustic data were collected. It is likely that Hansbreen accelerated in September due to heavy rainfall events; such 'autumn acceleration' has been reported for other tidewater glaciers in Svalbard (e.g., Luckman et al., 2015; Schellenberger et al., 2015). Moreover, stake measurements do not capture ice velocity variations along the glacier terminus. The results of the experiment presented here are encouraging, given the limitations listed above and completely independent nature of the methods used to make the measurements.

## 7 Long-term Monitoring of Subaerial Calving Fluxes

The study of Glowacki and Deane (2020) demonstrated the potential for underwater ambient sound in glacial terminal bays to provide estimates of subaerial calving flux. Guided by this new opportunity, an impact-to-noise conversion model has been applied here to a roughly 1-month long continuous recording of ambient sound in the bay of Hansbreen in Svalbard to estimate subaerial calving flux. Good agreement is found between acoustic measurements of ice volume loss and the camera observations of the terminus area loss. The cumulative subarial calving flux estimated using the acoustic approach is higher than the ice loss derived using a pair of satellite images combined with stake measurements of the glacier velocity. Clearly, the acoustic method requires further improvements, but the results are encouraging, especially as acoustic recorders are increasingly deployed in the Arctic for other studies, such as studies of biodiversity or human impacts like shipping, and they can be harnessed to also provide measurements of any neighboring glaciers. This study reinforces the idea that passive underwater acoustics offers a new and viable method for monitoring mass loss from marine-terminating glaciers and ice shelves.

The development of algorithms for the automated detection of calving events and the removal of interfering events, such as disintegrating icebergs and flow noise, creates the possibility of monitoring the stability of tidewater glacier termini and ice shelves over long time scales. We envision a scenario where a number of relatively inexpensive autonomous underwater recording systems are deployed over year-long intervals around the termini of selected glaciers on a recurrent schedule. The analysis of the resulting signals would eventually allow glacial stability to be monitored over decadal and longer time scales. Concurrent measurements of relevant environmental drivers, such as water temperature, insolation and precipitation for example, could provide important insights into the processes controlling terminus ablation.

Personal experience making long-term recordings in the Arctic has taught us that not all deployed recording systems are recovered. It is therefore essential to keep the unit cost of recording systems low to allow for as much redundancy as possible within a fixed budget. Power consumption and data storage are two important drivers of cost. Requirements for both of these factors can be reduced by recording ambient sound on a fixed schedule or recording whenever the sound level exceeds a preset threshold. These two possibilities will now be discussed.

Figure 4 shows the effect of recording coverage on the estimated total number of calving events (A) and cumulative ice volume loss (B) over the study period. A continuous acoustic record corresponds to $\Delta = 0\%$. Periods of active recording change from 1 hour per day ($\approx 4\%$) to full coverage. Final estimates are derived proportionally, i.e. values obtained for $10\%$ and $20\%$ coverage are multiplied by 10 and 5, respectively. The specific hours of recordings during a day are selected repeatedly



using all possible combinations, creating a distribution of outputs for the chosen value of coverage. The event count is much less sensitive to the recording schedule than the subaerial calving flux. Nevertheless, one standard deviation of the estimated ice volume loss at 30% coverage is within 10% of the reference value. This result demonstrates the possibility of significant reduction in recording time (by 70%) with relatively low increase in the resulting error.

  Figure 5 shows the percentage of time occupied by calving noise during the experiment and the acoustic estimate of daily
calving rate. On average, calving events were active for only about 1% of the time; this corresponds to around 120 events per day. However, we expect that the calving detector may split single calving events into several detections. This is especially likely in the case of free falls of highly disintegrated icebergs. Therefore, we assume that the real daily calving rate is likely lower than the acoustic estimate. Although calving activity changes significantly with seasons and geographical location, it is still expected to be a small fraction of the total record. Exploiting this fact through the use of event-driven recording systems
could result in significant power savings for autonomous recorders.

## 8 Concluding Remarks

This first automated analysis of underwater sound recorded near the terminus of a tidewater glacier, compared with camera and satellite observations, demonstrates the feasibility of using cryoacoustics to monitor subaerial calving fluxes over extended periods. Challenges remain. The method has been verified for Hansbreen, a glacier of a scale common to Spitsbergen, but the
260 glaciers in Greenland can be factors of 10 or more larger in both horizontal and vertical scales. Moreover, calving mode and propagation conditions will vary between glaciers and there will be variability in levels of ice coverage in the terminus bay, which depend on calving rate, circulatory flow (long, narrow fjords are more prone to melange buildup), and sea ice coverage. Some or all of these factors may need to be accounted for when applying cryoacoustics to quantify subaerial calving flux between tidewater glaciers.
Cryoacoustics meets some of the important requirements for long-term monitoring of subaerial calving fluxes. The high rate of signal acquisition ensures that every calving event can be detected, provided its signal is above the detection threshold and below the rejection threshold. Although the uncertainty in the volume estimate for a single event is high, the total volume estimate becomes precise if a sufficient number of events are accumulated (Glowacki and Deane, 2020). With current technological limitations, it is not possible to take time-lapse photographs with high enough frame rates to ensure that all events
are captured. Moreover, cryoacoustics is insensitive to lighting conditions and relatively insensitive to weather conditions. For example, the dominant spectral component of noise generated by rain lies well above the 'calving band' (Pumphrey and Crum, 1988).

  Both cryoacoustics and cryoseismology provide temporally-resolved, continuous records of calving activity that are expected to be relatively insensitive to changing environmental conditions (Podolskiy and Walter, 2016; Deane et al., 2019).
Cryoacoustics has the additional benefit that the physics of sound production, which stems from the entrainment of bubbles around falling blocks of ice and collective oscillations of bubble plumes (Glowacki, 2020), is expected to apply broadly across calving glaciers in polar regions. If true, this will enable subaerial calving flux to be extracted from cryoacoustic signals with-



out the burden of a glacier-by-glacier calibration. Moreover, it may be possible to extract other geophysical signals of interest, such as terminus melt rate, from cryoacoustic records (Pettit et al., 2015; Deane et al., 2019).

Cryoacoustics, combined with other remote sensing methods, such as photogrammetry, cryoseismology and satellite observations, can form the core of a long-term monitoring system for subaerial calving fluxes from glaciers and ice shelves.

*Data availability.*   The acoustic data used in this study are available upon request from J.T. (jaroslaw.tegowski@ug.edu.pl)

## Appendix A: Materials and Methods

### A1   Acoustic and Camera Observations

Acoustic and camera data were collected around Hansbreen from 5 – 30 September, 2013. Noise recordings were made with a High Tech. Inc. HTI-96-MIN omnidirectional hydrophone mounted on the seafloor at a depth of 22 m and approximately 1.95 km from the glacier terminus. Data were sampled at a frequency of 32 kHz with 16 bits of dynamic range. Photographs of the terminus were taken every 3 hours using a Canon D1000 time-lapse camera located 160 m above sea level and 0.8 to 2 km from the glacier terminus (see Fig. 1).

The images were analyzed using a differences technique to estimate the area of ice lost from the cliff face through calving when lighting and weather conditions permitted. Changes at the calving front associated with calving events were found by comparing two consecutive time-lapse images. The terminus area loss was then estimated using known ice cliff heights at given locations. This methodology is sensitive to the variability of the terminus shape along its edge due to the oblique angle of the camera view. Moreover, we are aware that the glacier can calve more than once at the same location during a 3-hour time

frame. Nevertheless, the image analysis provides useful information about the overall calving activity at Hansbreen during the study period.

### A2   Satellite Observations

The volume of ice lost from Hansbreen through subaerial calving was estimated from satellite imagery and GPS observations of the glacier surface movement. subaerial calving flux is estimated using $Q = \bar{U}_{ice} + U_{front}LH$, where $Q$ is the volumetric

flux of icebergs, $\bar{U}_{ice}$ is the mean ice flow velocity, $U_{front}$ is the front retreat/advance velocity, and $L$ and $H$ respectively are the cliff length and height above sea level. The length of Hansbreen's active cliff and the average front retreat per day have been extracted from two multispectral Landsat 8 satellite images (2013-08-24 and 2013-09-25). Cliff height above sea level was determined in 2015 using a Riegl VZ-6000 high resolution 3D laser scanner. The glacier surface velocity was estimated using a mass balance stake mounted near the calving front (designated GPS in Fig. A1). Changes in stake position were

measured with precise dGPS in August 2013. The subaerial calving flux estimated over the 26 days of acoustic measurements is $(7 \pm 2) \times 10^6$ m$^3$, i.e. ca. $0.27 \times 10^6$ m$^3$ d$^{-1}$. This is the result used for comparison with the acoustic data. As a crosscheck,





the glacier surface velocity field was estimated using offset tracking from TerraSAR satellite radar images (2012-12-15 and 2012-12-26). subaerial calving flux was then estimated within 40 m sections of the glacier front to allow for variations in velocity and cliff height (see the grey profile in Fig. A1). The total volume of icebergs calved during the observation period is estimated to be $(6 \pm 1) \times 10^6$ m$^3$. It should be borne in mind, however, that TerraSAR images were collected in December 2012 and not during the study period. Nevertheless, the results show good agreement with subaerial calving flux estimated using stake measurements.

### A3 Calving Impact Energy Estimation

The calving impact energy was estimated from the noise energy using a power-law relationship model shown in Eq. 4. The parameters of the model are based on the calibration dataset presented in Glowacki and Deane (2020) that includes time-lapse images and underwater noise recordings of 169 calving events observed in 2016 at Hansbreen. However, the power-law model proposed by Glowacki and Deane (2020) was improved through the re-analysis of the calibration dataset. The following changes/modifications were applied:

1. Subsequent to publication, it was discovered that the acoustic recordings contained a small DC offset, which created a minor bias in the event energy estimates. This issue has been corrected here by running the acoustic recordings through a (30 - 100) Hz band-pass filter, the lower frequency of which is roughly equal to the low-frequency cut-off of the waveguide comprised of the sea surface and seafloor of the terminus bay.

2. The Bellhop ray-tracing model was used in Glowacki and Deane (2020) to calculate the loss of acoustic energy between the iceberg/water impact locations and the hydrophone. Here, the transmission loss of the calving noise was calculated using the parabolic equation model RAM (Collins, 1993) that is more recommended for low-frequency problems (Jensen et al., 2011). The grain size of the sediments and source depth were set to $\phi = 5$ and $Z_s = 5$ m, respectively. The attenuation, sound speed and density of the sediments were calculated from the grain size using formulas proposed by Hamilton (1972) and Hamilton and Bachman (1982). Moreover, in the present study, transmission losses were evaluated for a whole range of frequencies between 30 and 100 Hz, while a single nominal source frequency of 50 Hz was used in Glowacki and Deane (2020).

3. In Glowacki and Deane (2020), the parameters of the power-law relationship between the impact energy and noise energy were estimated by performing a linear least-mean-squares analysis of log-transformed variables; the impact noise energy was used as a dependent variable to estimate the impact-to-noise conversion coefficients. Here, the block impact energy was used as a dependent variable to minimize the error in the subaerial calving flux. Moreover, the coefficients of the power-law relationship were estimated using a non-linear regression model and non-transformed variables (function *fitnlm* in Matlab).

Problems remain. Most importantly, the calibration dataset is limited to 169 calving events and certainly do not represent the full range of possible values of the impact noise energy at Hansbreen. As a consequence, the power-law model – that is based





on the calibration data – certainly requires further improvement. This is confirmed by Fig. A2, which compares histograms and
empirical cumulative distribution functions (ECDFs) of $E_{ac,imp}$ computed for acoustic data collected in 2013 (present study)
and 2016 (calibration dataset). There are two major conclusions from Fig. A2.

First, the 2016 data do not include impact noise energies smaller than 10 joules. This is to be expected because the calibration
dataset requires that all calving events are also captured by the time-lapse camera. Small-scale changes at the glacier terminus
were hardly visible on camera images and could not be included in the calibration dataset. Individually, low-magnitude calving
events contribute little to the total subaerial calving flux; however, it cannot be ruled out that sometimes they constitute a large
fraction of the calving inventory and are meaningful as a whole.

Second, the 2016 data do not include calving events with $E_{ac,imp} > 4000$ joules either. The lack of the largest events in the
calibration dataset may be due to several reasons: (i) Hansbreen may have been in different melting regimes in 2013 and 2016,
which caused differences in calving styles (e.g., differences in the water depth at the terminus, glacier surface velocity, water
content, etc.), (ii) co-occurrence of large-scale calving events and bad weather/lighting conditions that prevented estimation of
impact energy from time-lapse images in the calibration dataset, (iii) underestimated transmission loss for the acoustic data
collected in 2013 because of the different location of the hydrophone compared to 2016 leading to a different propagation path
(note the shallow sill close to the recorder in Fig. 1A). Leaving aside the causes, the lack of the highest impact noise energies in
the calibration dataset results in extrapolation of the power-law model in the analysis of data collected in 2013; unfortunately,
the extrapolation applies to calving events that contribute the most to the subaerial calving flux. This is one possible explanation
of the discrepancy between calving fluxes derived from traditional methods and the acoustic technique.

The improvement of the power-law model in Eq. 4 requires long-term acoustic measurements close to different glaciers
combined with measurements of ice block volumes with optical techniques (e.g., time-lapse photography, terrestrial laser
scanning). Collecting calibration data that would include a whole range of styles and magnitudes of calving events is critical
for the usefulness of the acoustic technique in the monitoring of subaerial calving fluxes.

## A4    Calving Event Detection Algorithm

The calving event detection algorithm depends on the selection of an event detection factor, $\beta$, and a frequency band over
which the ambient sound is filtered before integration over time.

Figure A3 shows a summary of how the number of detected calving events varies with the detection factor $\beta$; interfering
events were not removed in this analysis. As might be expected, the total number of events decreases rapidly with increasing
detection factor because increasing $\beta$ results in the exclusion of the lowest-energy events. The results presented in Fig. 3 were
generated using $\beta = 5$, which was found to be an optimal selection in terms of limiting detection of non-calving events and
minimizing the loss of true calving events. The selection of $\beta$ consisted of the visual analysis of noise spectrograms and audible
inspection of sound recordings performed for a number of detections.

The detection factor selected here for the terminal bay of Hansbreen is likely not the optimal detection factor for other
environments or geometries for the terminus and recording station. This is because the detector performance is sensitive to the



signal-to-noise ratio at the hydrophone location and activity of other sources, such as nearby icebergs, which will vary from site to site.

It is noted in the main text that the event detection algorithm is based on the assumption that a good estimate for the
375 background noise is the median noise level in the detector frequency band. This assumption holds well when calving events are relatively infrequent within an observational interval selected for analysis (10 minutes here) but may bias the detector toward the exclusion of low energy events if too many events occur within the interval. The validity of the assumption was tested by computing the background noise level iteratively: detected events are removed from the record and a baseline noise level recomputed in successive passes through the data segment. The results of this analysis for change in number of events detected,
total acoustic energy at the receiver and summed event durations are shown in Fig. A4. The total percentage change in acoustic energy of all detected events is less than 3%, which is not a significant source of error given other, larger uncertainties in the analysis, such as hydrophone calibration and propagation loss. The percentage change in number of events is similarly minor. There is a larger impact on the duration of events because of two effects: 1. increasing the baseline noise level increases the detection threshold and decreases event duration and 2. the division of a single event into 2 or more events is sensitive to the
baseline noise level.

Here we use a simple but robust algorithm for the detection of subaerial calving events. The calving detector in the present form has two major weaknesses: (1) subaerial and submarine calving events cannot be distinguished and treated separately and (2) certainly there are still some unexplored interfering events that either cause false detections or periodically lower the signal-to-noise ratio, which makes calving detection more difficult (i.e., some calving events remain undetected). The latter
issue can be partly addressed with the use of more sophisticated detection techniques. See, for example, recent work in the cryoseismology community by Carr et al. (2020) and Köhler et al. (2022). However, we speculate that better understanding of different (non-calving) sound-source mechanisms and their underwater acoustic signatures is required before more sophisticated algorithms are implemented. Examples may include but are not limited to interactions between floating growlers and icebergs, ice fracturing events, water outflows from subglacial conduits, coastline landslides, and activity of marine mammals.
Most of these "interfering" noise sources can be investigated with meteorological and oceanographic measurements, high-frequency photographic images, laser scans, or radar scans synchronized with acoustic recordings. For example, Deane et al. (2014) reported that interactions of surface gravity waves with underside of ice ledges at the periphery of icebergs are sources of underwater noise emission below 500 Hz. Some sound-source mechanisms, like underwater fracturing events, require novel observational techniques or scaled laboratory experiments.

The problem remains with detection of submarine events. An inclusion of submarine events would be beneficial for two reasons: (i) to find out how frequent these events are and (ii) to provide more accurate estimates of subaerial calving fluxes by rejecting submarine events from the analysis. Moreover, it should not be ruled out that the methodology for estimating submarine calving fluxes from sound recordings made in glacial bays could be developed. If so, it would be possible to derive both subaerial and submarine calving fluxes using acoustic techniques. Recent work by Glowacki (2022) demonstrated that
the two calving modes can be acoustically distinguished using parameters of the log-normal distribution of the calving noise combined with calving signal duration. A semi-automatic detection of start and end times of calving noise was used. However,



the problem remains of how to automatically select $t_0$ and $t_1$ for submarine calving events; this would require long-term calibration dataset of synchronized high-frequency time-lapse images and noise recordings. Such data are not available for calving events observed in 2013. Consequently, the analysis of submarine events was out of the scope of this study.

*Author contributions.* J.T. conceived the project, led the 2013 field expedition to Svalbard and participated in data analysis and interpretation, and participated in manuscript preparation. O.G. is co-architect of the sound inversion methodology, performed data analysis and interpretation, and participated in manuscript preparation. M.C. collected and processed the photographic data of terminus ablation. M.B. collected and processed satellite and stick data of glacier velocity and ice mass loss. J.J. provided scientific oversight into data analysis and participated in manuscript preparation. M.M. participated in the field campaigns and provided CTD data. P.B. participated in field campaigns

and participated in manuscript preparation. G.B.D. is co-architect of the sound inversion methodology and detection algorithm, participated in field campaign, and participated in manuscript preparation.

*Competing interests.* The authors declare that they have no conflict of interest.

*Acknowledgements.* This work has been supported by the National Science Centre, Poland (grants 2011/03/B/ST10/04275 and 2021/43/D/ST10/00616), Ministry of Science and Higher Education of Poland (grant 1621/MOB/V/2017/0 and subsidy for the Institute of Geophysics, Polish Academy

of Sciences), U.S. Office of Naval Research, Ocean Acoustics Division (grant N00014-17-1-2633), USA National Science Foundation Office of Polar Programs (grant OPP-1748265), and Research Council of Norway (*Arctic Field Grant*, RIS ID: 6133). We gratefully acknowledge the support of the staff at the Polish Polar Research Station in Hornsund (PPS). CTD data were collected under the oceanographic monitoring of the PPS (available at https://dataportal.igf.edu.pl/). TerraSAR-X data provided by German Aerospace Center (DLR, project LAN2787). Landsat-8 data made available by the United States Geological Survey (USGS).



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



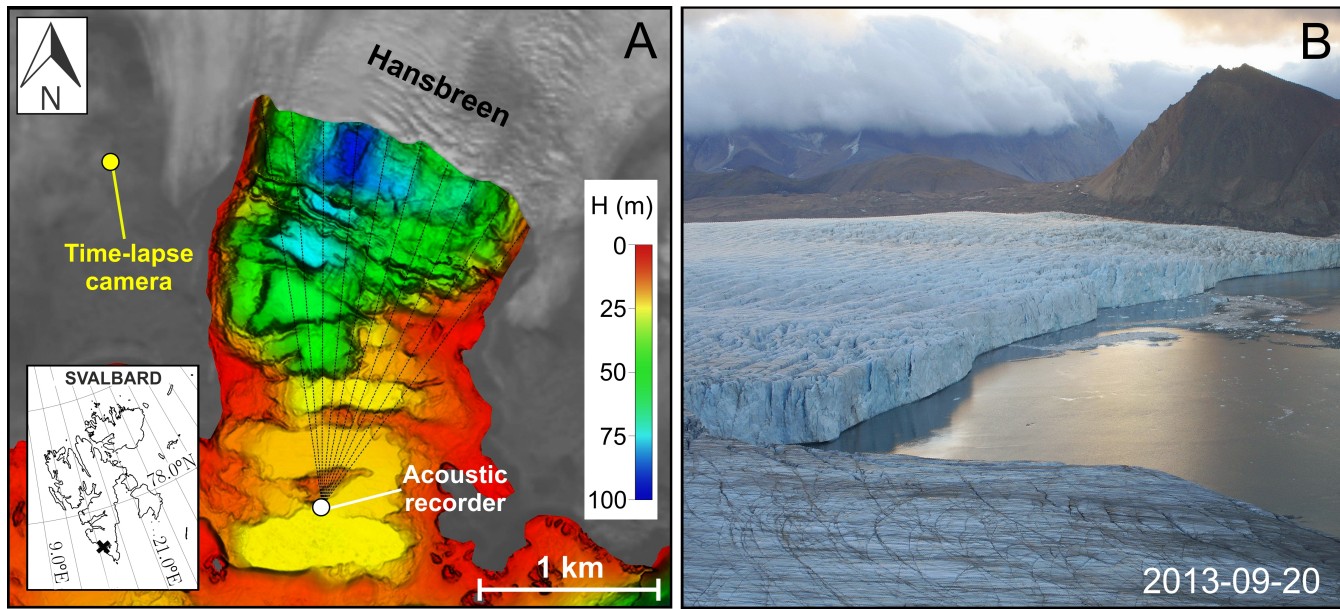

**Figure 1.** (A) Map of the study site and (B) example image of the Hansbreen terminus. Locations of the time-lapse camera and the acoustic buoy are marked with yellow and white font, respectively. The black dashed lines show different propagation transects that were used for estimating the noise transmission loss. Landsat 8 satellite data collected on 11 September 2013, courtesy of the US Geological Survey, Department of the Interior. Bathymetric data provided by (1) the Norwegian Hydrographic Service under the permit no. 13/G722, issued to the Institute of Geophysics Polish Academy of Sciences, and (2) the Faculty of Natural Sciences, Institute of Earth Sciences, University of Silesia in Katowice, Sosnowiec, Poland (Błaszczyk et al., 2021).

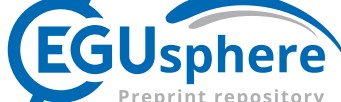

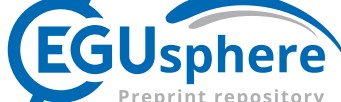

**Figure 2.** Calving detector explained. Top: A spectrogram of the sound produced by a calving event within the background noise radiated by the melting terminus of Hansbreen. The frequencies $f_0$ and $f_1$ on the right hand side of the plot denote the lower and upper limits of the detection frequencies (see text for details). Bottom: Noise power from the spectrogram above, integrated from $f_0$ to $f_1$ and filtered. The calving event is detected when the noise power exceeds the power threshold, $P_{thres}$, as annotated by $t_0$ and $t_1$. The threshold power is based on the baseline power, $P_{base}$, which is determined from a statistical analysis of the sound (see text for details).



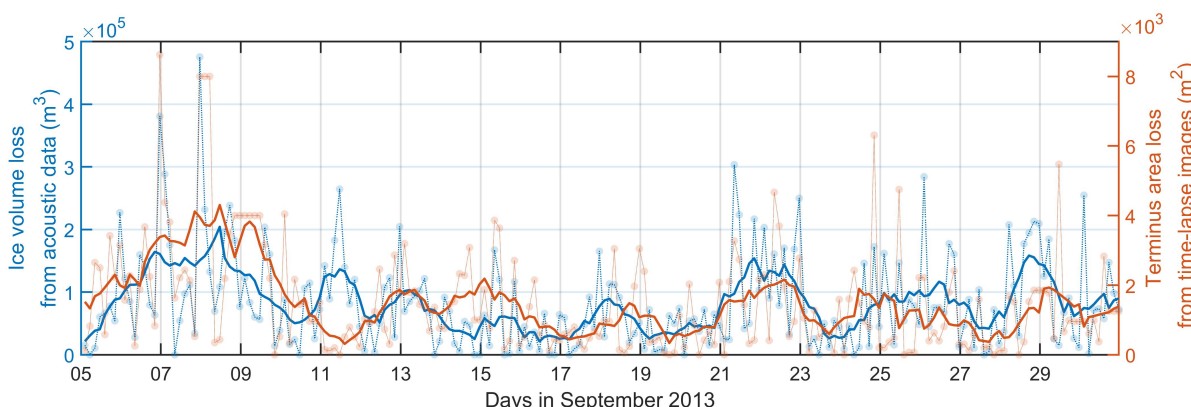

**Figure 3.** A comparison of ice volume loss derived from iceberg impact noise and terminus area loss from camera observations. The ice volume loss was calculated every 3 hours, which is the sampling time of the camera system. The thick lines show the two data streams smoothed with 1-day running average.



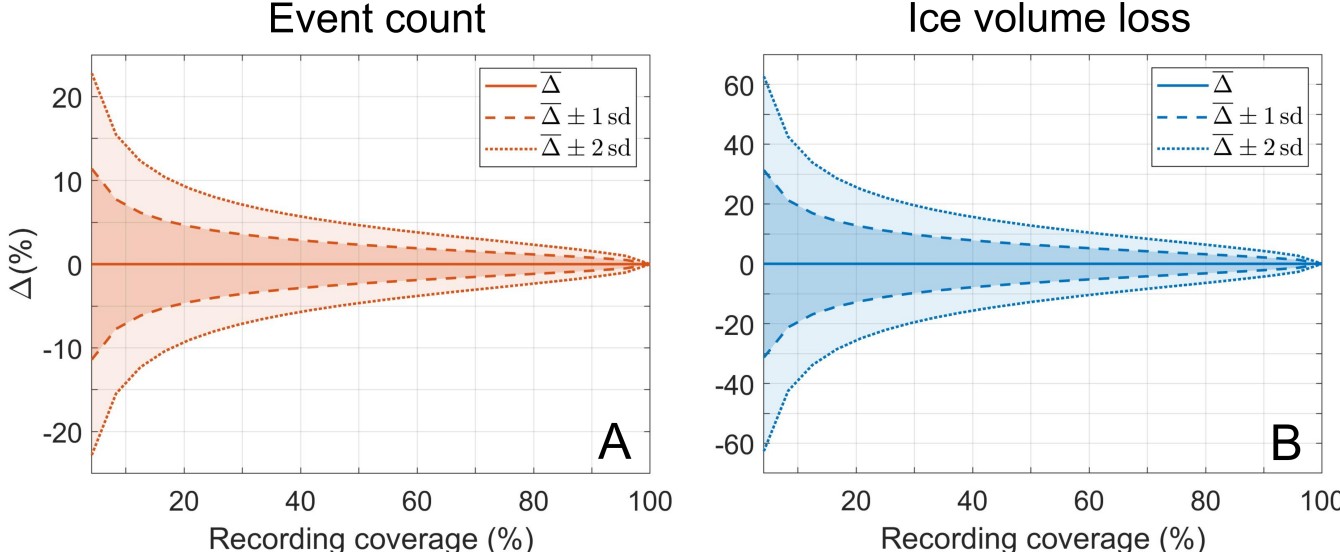

**Figure 4.** The effect of sampling coverage on acoustic estimates of (A) event count and (B) ice volume loss. Recording periods range from 1 hour per day to continuous. The reference value of $\Delta = 0\%$ is for full coverage.



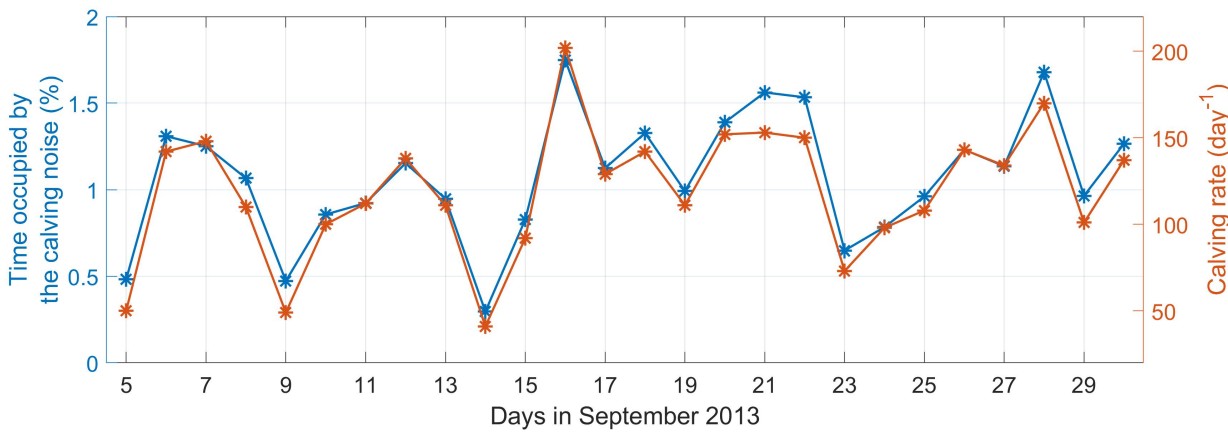

**Figure 5.** (blue) Percentage of time occupied by calving noise over the study period and (orange) the acoustic estimate of daily calving rate.





**Figure A1.** Data used in the subaerial calving flux calculations. Horizontal velocity map from offset tracking on repeat TerraSAR satellite radar images (2012-12-15 and 2012-12-26), retreat of Hansbreen cliff and localization of stake position with dGPS measurements between August 5 and August 24, 2013. The grey profile with dots shows division of the ice cliff into 40 m lengths, along which precise calculation of subaerial calving flux were made.





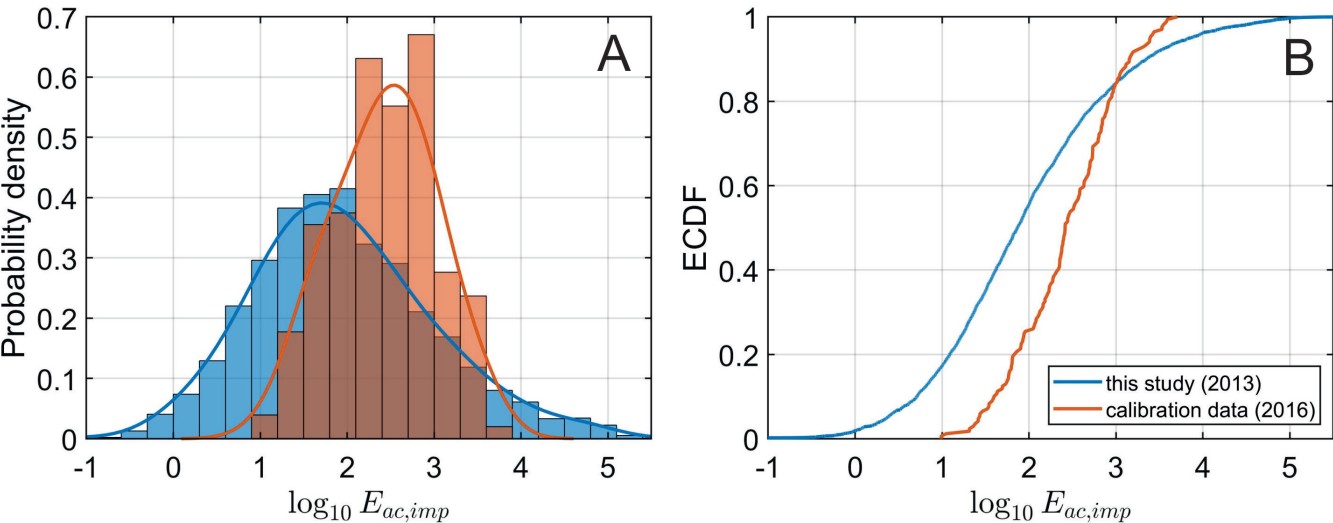

**Figure A2.** (A) Histograms and (B) empirical cumulative distribution functions (ECDFs) of log-transformed impact noise energies calculated for calibration data (2016; orange) and acoustic measurements used in this study (2013; blue).



**Figure A3.** Relationship between detection factor $\beta$ and the number of detected calving events. Interfering events are not removed in this analysis.

**Figure A4.** Change in number of events detected (N), total acoustic energy at the receiver (E) and summed event durations (D) versus iteration number for the background noise estimate.