# Peer review of "Monitoring Glacier Calving using Underwater Sound"

_EGUsphere, 2023_

## Author Comment (AC1)

Dear Dr. Podolskiy,

thank you for your insightful comments and suggestions. We will address all minor recommendations as suggested in the revised version of the manuscript. Please find below our responses to your more moderate remarks.

I feel there is some incoherency ("oranges / apples") in comparison of sound-retrieved volume to geometric changes. Specifically,

- camera data was used for vertical area estimate, without conversion to volume.

- satellite data was used for area estimate with conversion to volume.

In both cases, one has to assume some weighting for going from 2D to 3D. For satellite data, this was cliff height measured in 2015. For camera data, a similar step was not attempted, while commonly used in papers, including those by the authors. Homogeneous comparison makes more sense to me, especially than satellite analysis is inevitably pretty rough and dGPS data are not relevant to the period of acoustic measurements.

We do agree that the comparison between the area loss and volume loss may introduce some confusion. Thank you for pointing this out. However, the purpose of Fig. 3 was to demonstrate the general strong correspondence of calving activity at Hansbreen determined from time-lapse images and underwater noise recordings. Computing glacier volume loss due to calving from the time-lapse images taken in the study period is problematic for two major reasons: (1) highly oblique camera view (see Fig. 1B in the manuscript) and (2) long time intervals between consecutive images (3 hours).  The latter can easily lead to the classification of several low-magnitude calving events originating in the same sector of the calving front as a single big event. In a such case, the volume loss calculated from the area loss would be largely overestimated. This is because the ice volume loss ($V$) is typically estimated from the newly exposed terminus area ($A$) using $V = C\,A^{3/4}$. We decided to stop the image analysis at the calculation of the terminus area loss to better represent the real calving activity. Nevertheless, we understand that more explanations are needed in the main text to justify our choice. We will address this issue in the revised version of the manuscript.

With due respect to the authors, "upon request" is not in line with open data policy of TC. Also, from personal experience, I got no replies to my two last requests for data to authors using similar statements or saw that email addresses became obsolete.

Agreed, we will share all data used in this study in public repositories.

Furthermore, I suppose by 2015-laser measurements, the glacier thinned and thus the ice cliff (i.e. cross-section) was lower than in 2013, which could lead to an underestimate of calving flux and could be checked.

We understand these concerns. However, Błaszczyk et al. (2021, https://doi.org/10.1029/2020JF005763, Fig. 4) showed that the average front height did not change significantly between 1991 and 2015, even when the glacier retreated to deeper water. We took into account the variability of the average terminus height by assuming its accuracy as ±5m.

"precise dGPS in August 2013"

Please indicate the exact dates, because as I understand, they were outside the hydro-acoustic monitoring period and their "precise" nature was of arguable help?

and also:

Please note that "stake measurements" were not shown and, if I am not missing something, the mean daily ice flow velocity was also not mentioned (Section 6). Considering high sensitivity of satellite-derived flux estimates to this value, this detail is important. It alone could explain the discrepancy between the acoustic and satellite derived fluxes.

Thank you for pointing this out. Changes in stake positions were measured with dGPS in August 6 and August 28. Indeed, these measurements were conducted outside the hydro-acoustic monitoring period and should be considered with caution. Moreover, we understand that measurements of a single stake is a very rough estimate that does not take into account the variability of glacier velocity along its calving front. However, this is the best approximation available that helps to compare the total ice volume loss estimated from satellite and acoustic data. The average ice velocity was 257 ma$^{-1}$. In the revised version of the manuscript we will also provide the initial and final stake coordinates in UTM coordinate system: (1) 515877.563, 8549150.693 and (2) 515872.566, 8549136.029. We will also modify the caption of Fig. A1 to make it more explanatory.

Furthermore, I was puzzled by "a cross-check" of velocity field by using December 2012 imagery, which tells us little about dynamics in the end of summer. Why December? As I understand, there are plenty of Glacier Image Velocimetry open-source tools to retrieve velocity field at good temporal resolution (e.g., https://doi.org/10.5194/tc-15-2115-2021)

We agree that there are different tools and ready-to-use velocity data, e.g. Itslive (https://itslive-dashboard.labs.nsidc.org/). However, in 2013, there still was not a lot of cloudless multispectral data, especially for the area of the Hornsund fiord. Only one from two Landsat 8 images that we used to determine terminus position was cloudless, so it was not possible to estimate velocity of the glacier.

The velocity measured in August with dGPS at a distance of roughly 200 m from the glacier terminus was 257 ma$^{-1}$. The average glacier velocity from TerraSAR-X velocity field in December 2012 at the same location where GPS measurement were conducted was 199 ma$^{-1}$. That gives the ratio of winter to summer velocity of 77%. Similar velocity ratios were found in other years (Błaszczyk et al. 2019, http://dx.doi.org/10.33265/polar.v38.3506). We could therefore speculate that the frontal ablation is underestimated by around 20-25% when using the ice velocity from winter. However, we also admit that the velocity ratio could be lower closer to the glacier terminus. This would bring the total

volume loss estimated from the satellite data closer to the results obtained with the acoustic method.

There are quite many alternative hypothesis for discrepancy between acoustic and image-derived fluxes, due to uncertainty in velocity, calibration, and etc. Perhaps reducing this number might be possible by estimating volume from photos and revisiting ice velocity?

Please see our previous comments. We believe there is no more room for improvement in terms of the accuracy of calving flux estimates. Addressing your remark would require new data that we do not have for the study period and location considered.

Line 290

to me, there is insufficient amount of detail, because it is not clear what is "a difference technique" and what exactly is compared (greyscale intensity?). Please elaborate because retrieval of area from oblique images is not a trivial task and the reader has little idea how to reproduce this. Please also see my comment on area vs. volume.

Agreed, we will clarify the image analysis in the revised version of the manuscript.

---

## Author Response (AR1)

Dear Editor

Dear Reviewers

Thank you again for all valuable comments. They were very helpful. Please find a revised manuscript and a marked-up manuscript version showing the changes made. We have revised the manuscript according to our previous responses in the discussion.

---

## Author Response (AR2)

Dear Editor

Dear Reviewers

We would like to again thank the reviewers for all their comments and suggestions. We do agree that our previous response to the reviewers' comments was inadequate. Please accept our apologies. Below we provide a detailed point-by-point response to all comments given by both reviewers. W hope that this time it will be much easier to follow our responses and corrections to the manuscript.

**Reviewer 1 – prof. Evgeny Podolskiy**

Lines 23, 32
it is my personal feeling, but I would keep "urgent" out of the manuscript.

We decided to keep this phrase as a legitimate one to emphasize the strong requirement for long-term observations.

Line 45
wind and precipitation examples need references.

Agreed. We provided two example references. See line 48 in the corrected manuscript: <<(e.g., Nystuen, 1986; Vagle et al., 1990)>>

Line 94
sound is {analyzed} into a spectrogram
-> transformed?

Corrected as suggested.

Line 128
can appear to be very large
->can be confused with very large?

Corrected as suggested.

Line 132
I suggest to explain what is alpha-stable distribution, since not many readers may follow.

Agreed. See new sentence in lines 136-137 of the corrected manuscript: < This should not come as a surprise as stable distributions allow skewness and heavy tails (Nolan, 2021).> Moreover, we provided some additional explanation for the alpha-parameter; see lines 137-138: <<- a parameter that determines the heaviness of the tails ->>.

Line 137
I would just mention that remote icebergs, say at the same distance as the calving front, but behind the receiver still remain an open problem.

Absolutely true! Thank you for this comment. See new sentences in lines 142-146 of the corrected manuscript: << It should be borne in mind, however, that the disintegration of icebergs located at a similar distance from the hydrophone as the glacier terminus can still be confused with calving events. This issue cannot be addressed in the present study because of the limited view and low temporal resolution of the time-lapse camera and inability to estimate the noise directionality (single-channel recordings). In future studies, two or more hydrophones could be used to identify and discard noise signals not related to the calving activity.>>

Lines 191-203
I feel there is some incoherency ("oranges / apples") in comparison of sound-retrieved volume to geometric changes. Specifically,

- camera data was used for vertical area estimate, without conversion to volume.

- satellite data was used for area estimate with conversion to volume.

In both cases, one has to assume some weighting for going from 2D to 3D. For satellite data, this was cliff height measured in 2015. For camera data, a similar step was not attempted, while commonly used in papers, including those by the authors. Homogeneous comparison makes more sense to me, especially than satellite analysis is inevitably pretty rough and dGPS data are not relevant to the period of acoustic measurements.

Following our previous response to this comment, we provided new explanations. Please see lines 211-218 in the corrected manuscript: << We understand that comparing terminus area loss from images and ice volume loss from acoustic recordings may introduce some confusion. Nevertheless, the purpose of this exercise was to demonstrate the general strong correspondence of calving activity at Hansbreen determined from time-lapse images and underwater noise recordings.
Computing glacier volume loss due to calving from time-lapse images taken in the study period is problematic for two major reasons: highly oblique camera view and relatively long time intervals between consecutive images (3 hours). The latter can easily lead to the classification of several low-magnitude calving events originating in the same sector of the calving front as a single big event. In a such case, the volume loss calculated from the area loss would be largely overestimated; therefore, to better represent the real calving activity, we have decided to stop the image analysis at the calculation of the terminus area loss.

Lines 204-217

There are quite many alternative hypothesis for discrepancy between acoustic and image-derived fluxes, due to uncertainty in velocity, calibration, and etc. Perhaps reducing this number might be possible by estimating volume from photos and revisiting ice velocity?

As we stated in our preliminary response: we believe there is no more room for improvement in terms of the accuracy of calving flux estimates. Addressing your remark would require new data that we do not have for the study period and location considered.

Furthermore, I suppose by 2015-laser measurements, the glacier thinned and thus the ice cliff (i.e. cross-section) was lower than in 2013, which could lead to an underestimate of calving flux and could be checked.

As we responded previously: we understand these concerns. However, Błaszczyk et al. (2021, https://doi.org/10.1029/2020JF005763, Fig. 4) showed that the average front height did not change significantly between 1991 and 2015, even when the glacier retreated to deeper water. We took into account the variability of the average terminus height by assuming its accuracy as ±5m.

To make it clear we added some explanation. Please see lines 328-329 in the corrected manuscript: <<Błaszczyk et al. (2021) showed that the average front height has not changed significantly between 1991 and 2015.>>

Data availability: (sound, dGPS, etc)
With due respect to the authors, "upon request" is not in line with open data policy of TC. Also, from personal experience, I got no replies to my two last requests for data to authors using similar statements or saw that email addresses became obsolete.

Agreed. All data that we used are now freely available. Please see a new data availability statement:

Data availability. The acoustic data used in this study are available at https://doi.org/10.34808/jp25-2b47. Time-lapse images of the Hansbreen's terminus can be downloaded from https://polaris.us.edu.pl/share.cgi?ssid=d62f7499b8b341ed9cb392e827367ad1. CTD data – used for noise propagation modeling – are available at https://dataportal.igf.edu.pl/dataset/inter-calibrated-temperature-and-salinity-in-depth-profilesin-hornsund-fjord.

Considering that the amount of sound data from polar regions is expected to increase, the code used for this paper might be helpful for the community and would increase citations to this work, so I encourage the authors to keep such code alive, say at github.

Agreed. However, we decided to choose a bit different strategy here: every step of the event detection algorithm is described in the text, with no shortcuts. The same applies to the calculation of the acoustic energy and removal of interfering events. This way it should be straightforward for the readers to (i) reproduce the results and (ii) play with their own data using our methodology. We believe that this is a good introduction of the technique for more general polar community. Nevertheless, we believe that this topic deserves a separate method paper that will include careful validation of the detection and removal algorithms based on high-frequency time-lapse images taken over long time period, for example. Then, the code could be shared with detailed sensitivity analysis. Unfortunately, this requires new data.

Line 290
to me, there is insufficient amount of detail, because it is not clear what is "a difference technique" and what exactly is compared (greyscale intensity?). Please elaborate because retrieval of area from oblique images is not a trivial task and the reader has little idea how to reproduce this. Please also see my comment on area vs. volume.

Agreed. First of all, the phrase "a difference technique" was itself confusing. We removed this phrase and added some new explanation. Please see line 317 in the corrected manuscript: <<(…) in terms of shape and color of the terminus area>>.

Line 299
Q = U_i + U_f*L*H
= m3/d ? please check units

Thank you for pointing this out. There was a missing bracket and a wrong sign. Please see the corrected equation: Q = (U_i - U_f)*L*H

Lines 299, 308
. subaerial calving flux
. Subaerial …

Corrected as suggested.

Line 305
"precise dGPS in August 2013"

Please indicate the exact dates, because as I understand, they were outside the hydro-acoustic monitoring period and their "precise" nature was of arguable help?

and

Line 312
Please note that "stake measurements" were not shown and, if I am not missing something, the mean daily ice flow velocity was also not mentioned (Section 6). Considering high sensitivity

of satellite-derived flux estimates to this value, this detail is important. It alone could explain the discrepancy between the acoustic and satellite derived fluxes.

Agreed. Please see lines 329- 336 in the corrected manuscript: << Changes in stake positions were measured with dGPS on August 6 and August 28, 2013. The initial and final stake coordinates in UTM coordinate system were: 515877.56 E, 8549150.69 N and 515872.57 E, 8549136.03 N. The average ice velocity was 257 ma-1.  We are aware that these measurements were conducted outside the hydro-acoustic monitoring period. Moreover, measurements with a single stake is a very rough estimate that does not take into account the variability of glacier velocity along its calving front. However, this is the best approximation available that helps to compare the total ice volume loss estimated from satellite and acoustic data. >>

Line 306
Furthermore, I was puzzled by "a cross-check" of velocity field by using December 2012 imagery, which tells us little about dynamics in the end of summer. Why December? As I understand, there are plenty of Glacier Image Velocimetry open-source tools to retrieve velocity field at good temporal resolution (e.g., https://doi.org/10.5194/tc-15-2115-2021)

As we stated in our preliminary response: we agree that there are different tools and ready-to-use velocity data, e.g. Itslive (https://itslive-dashboard.labs.nsidc.org/). However, in 2013, there still was not a lot of cloudless multispectral data, especially for the area of the Hornsund fiord. Only one from two Landsat 8 images that we used to determine terminus position was cloudless, so it was not possible to estimate velocity of the glacier.
The velocity measured in August with dGPS at a distance of roughly 200 m from the glacier terminus was 257 ma-1. The average glacier velocity from TerraSAR-X velocity field in December 2012 at the same location where GPS measurement were conducted was 199 ma-1. That gives the ratio of winter to summer velocity of 77%. Similar velocity ratios were found in other years (Błaszczyk et al. 2019, http://dx.doi.org/10.33265/polar.v38.3506). We could therefore speculate that the frontal ablation is underestimated by around 20-25% when using the ice velocity from winter. However, we also admit that the velocity ratio could be lower closer to the glacier terminus. This would bring the total volume loss estimated from the satellite data closer to the results obtained with the acoustic method.

We added information about the lack of cloudless multispectral data covering the study period. See lines 341-342 in the corrected manuscript: << There is a lack of cloudless multispectral data from 2013 for the area of the Hornsund fiord. >>.

Line 326
please show units for grain size

Please note that phi scale for grain size is non-dimensional. To make everything more clear, we added the corresponding diameter in μm: <<(diameter D = 31μm)>>. Hope that helps.

**Reviewer 2**

The abstract and introduction present information that sets the scene and justifies the interest in glacial loss, relying quite heavily on sea level rise aspects for this. This is fair enough, but there are also dynamical implications from calving glaciers that add to such drivers, and which are separate from sea level rise considerations. These include ocean mixing, and things that depend on that (e.g. productivity, climate, sea ice production). Can a few sentences of text be included to explain this importance?

Good point. Please see two additional sentences added in lines 37-39 of the corrected manuscript: << Calving is responsible not only for ice loss itself but also for profound changes in ocean mixing, which affect sea ice formation and marine productivity. Meredith et al. (2022) have recently demonstrated that internal tsunamis triggered by calving events are important drivers of regional shelf mixing. >> Moreover, we added <<ocean mixing>> in the second line of the abstract and << freshening of the ocean combined with>> in line 22 of the introduction.

Abstract and Introduction discuss the Arctic and Greenland in the context of glacial change, with good justification. Some other regions also matter in this context, e.g. Antarctic Peninsula, Patagonia etc. Perhaps mentioning these would broaden the applicability even further in the readers' mind?

Agreed. To address this, we made some corrections and added new text:

1. In the first line of the abstract we changed "the Arctic" to <<glaciated areas>> .
2. See lines 22-23 in the corrected manuscript: <<Glaciers and ice caps in Patagonia, Svalbard, Antarctic Peninsula and other glaciated regions are also losing mass at an accelerated pace (IPCC, 2021).>>

Line 16. RCP8.5 is looking very unlikely as a trajectory that we are likely to follow. Perhaps include numbers for e.g. RCP4.5 instead, if available?

Agreed. However, we could not find numbers for RCP4.5. Instead, we decided to change RCP scenarios to socio-economic development scenarios (SSPs), which seem to be more reasonable. See lines 15-17 in the corrected manuscript: << Greenland's contribution to sea level rise by the end of the 21$^{st}$ century relative to 1995–2014 is estimated to be 0.06 (0.01 to 0.10) m and 0.08 (0.04 to 0.13) m under SSP1-2.6 and SSP2-4.5 socio-economic development scenarios, respectively (IPCC, 2021); these estimates exclude peripheral glaciers and ice caps. >>.

Line 25. "plunged into darkness" sounds a bit dramatic - perhaps just "subject to darkness"?

Corrected as suggested.

Line 63. Figure 1 is not really a schematic, but instead a map and a photograph (both of which are good).

Agreed, see corrected text in line 66: << A map of the study site and an example photograph of the terminus of Hansbreen are shown in Fig. 1.>>

Line 71. Salinity, as measured, is a ratio and hence is dimensionless. While one sees "PSU" quite often, it is not actually correct. Correct terminology would be "... a salinity of 30 on the practical salinity scale".

Agreed, please see the modified text in lines 73-74 of the corrected manuscript: << The water temperature and salinity **(on a practical scale)** in the center of the bay ranged from -1.8 °C to more than 2.0 °C and from **30 to almost 35** during 2015 and 2016 **(Moskalik et al., 2018).** >>. Also: please note that a reference is added to a paper that describes temperature and salinity profiles in the study site. Moreover, the data availability statement provide information hot to download CTD data collected under the long-term oceanographic monitoring in Hornsund.

Line 105 and about. It would be worth some text explaining the extent to which the choice of constants and thresholds are likely specific to the location under study - this has relevance when considering how to apply this technique at other locations.

Agreed. See lines 111-113 in the corrected manuscript: << It should be borne in mind, however, that different study sites and experiment geometries may require different constants and thresholds for effective calving detection (see Appendix A: Materials and Methods).>>

Please also note that this topic is discussed in Appendix A. See, for example, lines 401-404: "The detection factor selected here for the terminal bay of Hansbreen is likely not the optimal detection factor for other environments or geometries for the terminus and recording station. This is because the detector performance is sensitive to the signal-to-noise ratio at the hydrophone location and activity of other sources, such as nearby icebergs, which will vary from site to site.".

Line 150. It seems to me that a big next step, if possible, would be automated algorithm for submarine calving detection and flux calculation. What is needed for that to be developed?

This topic is discussed in section A4 of the Appendix. To make it clear for the reader, we added in lines 160-162: << The analysis of calving flux from submarine events would first require new calibration data (see section A4 in Appendix A for more details). >>

Line 177. Units are here written as "decibels" and were "dB" previously - I personally dont mind which, but need to be consistent.

Agreed. We decided to use "dB".

Line 232. This is a good idea, but how would the glaciers be selected? It is not possible to monitor all - can the most representative be chosen somehow? What would be the criteria for this?

Good question. We elaborated this idea by adding << There are many possible selection criteria but we are interested in glaciers that contribute the most to the sea level rise and stability of major ice sheets >> in lines 250-252 of the corrected manuscript.

Line 234. It is a good idea to monitor concurrent environmental drivers; surely it would be a good idea to also monitor current environmental impacts, e.g. ocean stratification etc?

Yes, definitely. Thank you for pointing this out. Please see a new sentence in lines 254-256 of the corrected manuscript: << Moreover, long-term acoustic monitoring programs could also help to better understand the impact of calving events on ocean stratification, structure and functioning of marine ecosystems, and glacier dynamics itself. >>.

Line 259. I'm sure this method has applicability elsewhere and on larger spatial scales, as stated in the text. I would presume that this method has great utility for the calving of grounded marine-terminating glaciers, but that calving of ice shelves or floating ice tongues would be different, since these would be more similar to detachment of an already-floating section of ice without necessarily the same impact on the ocean? Might be worth making this explicit if so, to avoid confusion.

Agreed. Please see new sentence in lines 284-285 of the corrected manuscript: <<For example, the detachment of an already floating section of ice from an ice tongue would not necessarily have the same impact on the ocean as a subaerial calving event from a grounded terminus.>>

Appendix. Is it worth detailing the CTD methodology, a little?

Readers can access all available CTD data for Hornsund using the repository detailed in the Data Availability section. Procedures and methodologies regarding the CTD data collection are described there in metadata.

---

## Author Response (AR3)

Dear Prof. Olaf Eisen,

thank you for handling our manuscript. We have incorporated all of your final suggestions.